# Dihydrogen Bonding—Seen through the Eyes of Vibrational Spectroscopy

**DOI:** 10.3390/molecules28010263

**Published:** 2022-12-28

**Authors:** Marek Freindorf, Margaret McCutcheon, Nassim Beiranvand, Elfi Kraka

**Affiliations:** Computational and Theoretical Chemistry Group (CATCO), Department of Chemistry, Southern Methodist University, 3215 Daniel Ave, Dallas, TX 75275-0314, USA

**Keywords:** dihydrogen bonding, local vibrational mode analysis, blue/red shifts, hydride complexes, hydrogen storage

## Abstract

In this work, we analyzed five groups of different dihydrogen bonding interactions and hydrogen clusters with an H${}_{3}^{+}$ kernel utilizing the local vibrational mode theory, developed by our group, complemented with the Quantum Theory of Atoms–in–Molecules analysis to assess the strength and nature of the dihydrogen bonds in these systems. We could show that the intrinsic strength of the dihydrogen bonds investigated is primarily related to the protonic bond as opposed to the hydridic bond; thus, this should be the region of focus when designing dihydrogen bonded complexes with a particular strength. We could also show that the popular discussion of the blue/red shifts of dihydrogen bonding based on the normal mode frequencies is hampered from mode–mode coupling and that a blue/red shift discussion based on local mode frequencies is more meaningful. Based on the bond analysis of the H${}_{3}^{+}$(H${}_{2}$)${}_{n}$ systems, we conclude that the bond strength in these crystal–like structures makes them interesting for potential hydrogen storage applications.

## 1. Introduction

Hydrogen bonding (HB) is one of the most prominent non–covalent chemical interactions [1,2,3,4,5], being responsible for the shape and structure of biopolymers found in nature, such as proteins and nucleic acids [6,7,8], but also being a major player in transition metal catalysis [9], solid–state chemistry and materials science [10]. Generally, HB occurs between a positively charged hydrogen atom of a proton donor, and a lone pair of an electronegative atom, π electron cloud, or a transition metal center, representing the corresponding proton acceptor. However, in recent years, in addition to this *conventional* HB interaction, another intriguing HB type has been discussed, where the two atoms directly involved in the weak interaction are both hydrogen atoms [11,12,13,14]. This interaction was coined by Richardson [15] *dihydrogen bonding* (DHB will be used for dihydrogen bonds in this study). This describes the attraction between a protonic (H${}^{\delta +}$) and a hydridic (H${}^{\delta -}$) hydrogen atom, where the protonic H atom acts as the proton donor as in conventional HBs, and the hydridic H atom takes over the role of the proton acceptor [11].

Interaction energies of DHBs are generally between 1 and 7 kcal/mol, and because of their similarity to HBs, they are often treated as a subclass of HBs [16]. It has been suggested that DHBs are predominantly of electrostatic nature [16,17,18,19], accompanied in stronger DHBs with a substantial covalent contribution [16,20,21,22]; although, this has not been quantified so far. DHBs can influence molecular properties in the gas phase, in solution and in the solid state, yielding broad potential utilities in catalysis and materials sciences. In the solid state, DHBs show the unique ability to lose $H_{2}$, which could be utilized for the design of low–cost, high–capacity hydrogen storage materials [23,24,25]. In particular, a better understanding of the transformation from a weak H−H electrostatic interaction to a strong covalent bond of $H_{2}$ type could open new routes for technologies using hydrogen as an environmentally clean and efficient fuel.

The first experimental observation of the attractive interaction between two hydrogen atoms was reported in early 1930 by Zachariasen and Mooney [26] in the crystal structure of ammonium hypophosphite (${\mathrm{NH}}_{4}H_{2}{\mathrm{PO}}_{2}$), showing a close contact between the H atoms of the hypophosphite group and those of ammonium. More than 30 years later, Burg [27] reported the direct intermolecular interaction between the H atoms of the NH and ${\mathrm{BH}}_{3}$ groups in liquid ${\left({\mathrm{CH}}_{3}\right)}_{2}\mathrm{NH}-{\mathrm{HBH}}_{2}$, suggesting the presence of an H−H interaction comparable to that of a HB. This interaction was formally categorized as a true HB by Brown in the late 1960s [28,29,30] on the basis of the IR spectra of boron coordination compounds.

DHB is particularly attractive for molecular systems involving metal hydrides acting as proton acceptors. The first experimental evidence for the formation of a DBH involving a metal hydride was presented independently by Crabtree [31] and Morris [32]. Later, Crabtree et al. [33,34] investigated the H−H interaction in molecular complexes involving iridium hydrides, exploring the strength of this interaction with experimental and computational means. By using the Cambridge Crystallographic Database [35] and spectroscopic data of aminoboranes, Richardson et al. [15] and later Crabtree et at. [36] showed that the NH−HB distances and heats of formation are in the ranges of 1.7–2.2 Å and 3–7 kcal/mol, respectively, comparable to that of conventional HBs.

Over the years, DHB has been also the subject of many theoretical studies utilizing different levels of theoretical approaches. Comprehensive overviews can be found in Refs. [1,2,3,14,37,38] and studies cited therein. Furthermore, two new data sets have been recently reported for benchmarking [39]. The focus of these theoretical studies ranges from energy decomposition and bond energies [40,41,42], geometries [1,11], normal mode frequency shifts [9,43], studies of the topological features of the total electron density ρ(r) [44,45,46,47], and electron localization function (ELF) investigations [48,49] to the analysis of electrostatic potential [50,51,52], which led to a number of open questions. In particular, what is missing so far is a quantitative bond strength measure coupled with the assessment of the covalent character, which allows the classification of DHB as (i) weak H⋯H interactions of the van der Waals type, (ii) moderate H⋯H interactions mostly of electrostatic type, and (iii) strong H−H interactions with substantial covalent character.

Therefore, the focus of our study was to introduce a quantitative bond strength measure based on the local vibrational mode analysis (LMA), originally developed by Konkoli and Cremer [53,54,55,56,57], paired with Weinhold’s Natural Bond Orbital (NBO) population analysis [58,59,60], and Bader’s Quantum Theory of Atoms–In–Molecules (QTAIM) analysis [61,62,63,64], in order to obtain a deeper understanding of DHBs in general, in particular their strength, and to unravel similarities/differences compared to conventional HBs. Figure 1 shows the set of DHB complexes investigated in this study (organized in six specific groups) covering a broad spectrum of DHBs, ranging from weak interactions of electrostatic origin to strong interactions with covalent character. Group I includes hydrides interacting with simple organic neutral and cationic molecules as well intramolecular DHB in boron complexes, and Group II shows DHB in ammonia–borane complexes. DHB in Fe complexes are included in Group III, while DHB in noble gas complexes are involved in Group IV. We also reassessed the H−H interaction in aromatic hydrocarbons such as phenanthrene, dibenz[a]anthracene, and biphenyl [22,65] in Group V, where both hydrogen atoms are equally charged, which has been called the H−H *bond* in the literature, to contrast this situation from DHB [11], leading to some controversy [66]. Group VI of our study is made up of DHB in H_3_^+^(H${}_{2}$)${}_{n}$ hydrogen clusters.

The manuscript is structured in the following way: In the results and discussion part, general findings for Group I–Group V are summarized, followed by a more specific discussion of each individual group and the description of H–H interactions in H_3_^+^(H${}_{2}$)${}_{n}$ clusters (Group VI, Figure 1). In the computational methods section, a short summary of LMA is given, and the model chemistry applied and software utilized are described. In the final part, the conclusions and future aspects of this work are presented.

## 2. Results

### 2.1. Overall Trends

In the following, general trends found for Group I–Group VI members are discussed referring to the data collected in Figure 2, Figure 3, Figure 4, Figure 5, Figure 6, Figure 7, Figure 8 and Figure 9 and Table 1, Table 2 and Table 3. In addition, bond lengths *R* in Å, local mode force constants $k^{a}$ in mdyn/Å, local mode vibrational frequencies $\omega_{a}$ in cm${}^{-1}$, binding energies *E* in kcal/mol, electron densities at the bond critical points $\rho_{c}$ in e/Å${}^{3}$ and energy densities at bond critical points $H_{c}$ in Hartree/Å${}^{3}$ of protonic and hydridic parts of DHBs and DHBs for all complexes and references compounds are compiled in Appendix A.

*Strength of DHBs:*Figure 2a–c shows BSO n values for the DHBs, protonic and hydridic parts of DHBs; in Figure 2d the relationship between protonic and hydridic local mode force constants $k^{a}$ is shown; and in Figure 2e,f the pairwise correlation between local mode force constants $k^{a}$(Protonic), $k^{a}$(Hydridic) and $k^{a}$(DHB) is shown. BSO n (DHB) values range from 0.14 (**FeF2**) to 0.41 (**BeO4**), which is in a similar range as we previously found for HBs (BSO n = 0.135, N–H⋯F; BSO n = 0.33, N–H⋯N; see Ref. [67]), quantifying that they are both of comparable strength with the former being somewhat stronger. The protonic parts of DHBs ranging from 0.49 for **BeO3** to 0.99 for **CF** (being close to the FH reference value of BSO n = 1) are stronger than their hydridic counter parts with BSO n values of 0.41 for **NaO** and 0.86 for **Ar3**, respectively. One could argue that a weaker protonic part of DHB caused by charge transfer to the hydridic bond should lead to the strengthening of the latter and vice versus, resulting in a correlation between the strengths of protonic and hydridic parts of DHBs. As is obvious from Figure 2d, there is no such correlation between $k^{a}$(Protonic) and $k^{a}$(Hydridic). On the contrary, according to Figure 2d, there is some trend that weaker protonic parts of DHBs are paired with weaker hydridic bonds, as in the case of **BeO3** and that stronger protonic parts of DHBs are paired with stronger hydridic parts of DHBs, as in the case of **CF**. According to Figure 2e, there is also a tendency that weaker protonic parts of DHBs correspond to stronger DHBs, e.g., as found for **BeO3**, whereas stronger protonic parts of DHBs correspond to weaker DHBs, indicating an larger electron density transfer from a weak protonic part of DHB to DHB, that the donor bond transfers electron density to the DHB. However, as depicted in Figure 2f, this tendency is less pronounced for the hydridic parts of DHB.

Overall, these findings suggest that if one wants to design a DHB with a certain strength, the focus should be on modifying and fine–tuning the protonic rather than the hydridic part of DHB guided by the local mode force constants as a sensitive tool monitoring the changes in the bond strengths.

**Figure 2 molecules-28-00263-f002:**
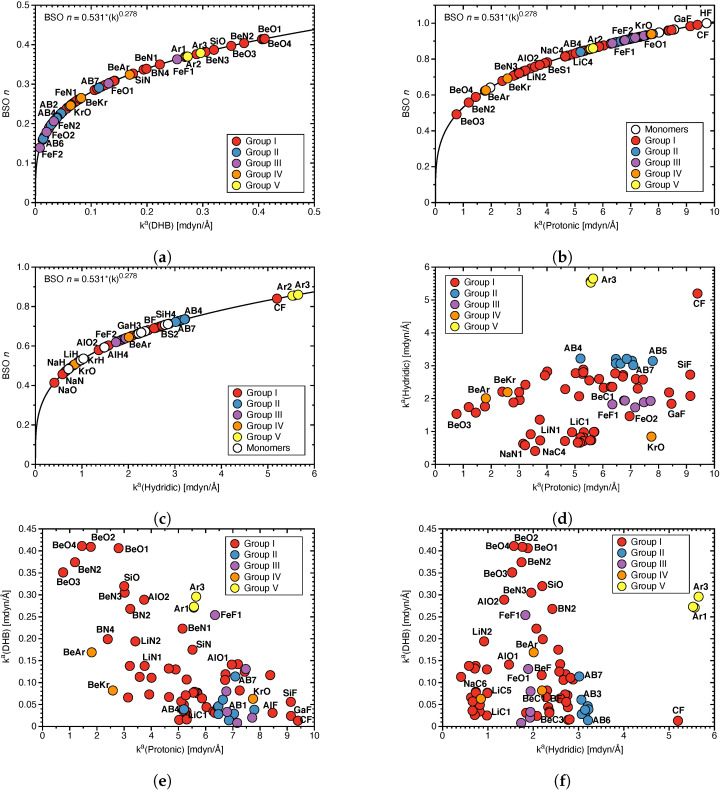
(**a**) BSO n values for DHB, (**b**) BSO n values for protonic parts of DHBs, and (**c**) BSO n values for hydridic parts of DHBs calculated from the corresponding local mode force constants as described in the text. (**d**) Correlation between $k^{a}$(Protonic) and $k^{a}$(Hydridic), (**e**) between $k^{a}$(Protonic) and $k^{a}$(DHB), (**f**) between $k^{a}$(Hydridic) and $k^{a}$(DHB).

**Figure 3 molecules-28-00263-f003:**
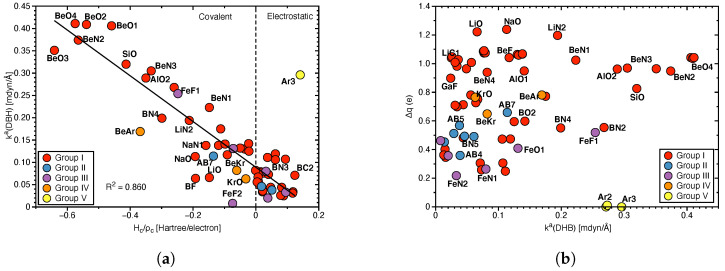
(**a**) Correlation of $k^{a}$(DHB) and normalized energy density $H_{c}/\rho_{c}$. (**b**) Correlation of charge differences Δq (see Table 1 for definition) and $k^{a}$(DHB).

*Covalency, protonic and hydridic charges:* The covalency of the DHBs investigated in this work was assessed by the normalized energy density $H_{c}/\rho_{c}$ (see Appendix A). In Figure 3a $k^{a}$(DHB) is correlated with $H_{c}/\rho_{c}$ which ranges from covalent $H_{c}/\rho_{c}$ = −0.6 Hartree/electron, to electrostatic $H_{c}/\rho_{c}$ = 0.2 Hartree/electron in accordance with the range of $H_{c}/\rho_{c}$ values −0.6 to 0.3 Hartree/electron for the HBs investigated in our previous work [67], disproving the assumption that DHBs are generally more electrostatic than their HB counterparts. Although there is no strong correlation (R${}^{2}$ = 0.860) between these two quantities, Figure 3a shows that stronger DHBs such as **BeO4** or **BeN2** are of covalent character whereas weak BDHs such as **BN3** or **BC2** are electrostatic in nature. In summary, the Cremer–Kraka criterion of covalent bond offers an efficient tool to assess the covalency of DHB interactions, which as our study shows can be quite different, depending on the electronic structure of the DHB complex and the nature of the protonic and hydridic parts of DHB.

In Table 1 and Table 2 protonic and hydridic H–atom NBO charges and their differences for Group I–V complexes are reported, and in Appendix A the NBO hydrogen charges for the reference compounds are shown.

**Table 1 molecules-28-00263-t001:** Protonic and hydridic NBO charges (in e) for Group I. Protonic hydrogen charge q(H${}^{\delta +}$); hydridic hydrogen charge q(H${}^{\delta -}$); charge difference Δq = q(H${}^{\delta +}$) − q(H${}^{\delta -}$).

Complex	q(H${}^{\delta +}$)	q(H${}^{\delta -}$)	Δq	Complex	q(H${}^{\delta +}$)	q(H${}^{\delta -}$)	Δq	Complex	q(H${}^{\delta +}$)	q(H${}^{\delta -}$)	Δq
LiC1	0.236	−0.804	1.040	BeN1	0.467	−0.556	1.024	BN5	0.404	−0.078	0.482
LiC2	0.213	−0.769	0.982	BeN2	0.418	−0.530	0.948	BO1	0.494	−0.104	0.598
LiC3	0.189	−0.776	0.964	BeN3	0.417	−0.552	0.969	BO2	0.491	−0.104	0.595
LiC4	0.249	−0.815	1.064	BeN4	0.439	−0.500	0.939	BS1	0.166	−0.083	0.249
LiC5	0.258	−0.816	1.074	BeO1	0.505	−0.537	1.042	BS2	0.166	−0.092	0.258
LiC6	0.269	−0.816	1.086	BeO2	0.521	−0.515	1.036	BF	0.565	−0.162	0.727
LiN1	0.326	−0.740	1.066	BeO3	0.497	−0.467	0.964	SiN	0.462	−0.310	0.772
LiN2	0.428	−0.769	1.197	BeO4	0.558	−0.484	1.042	SiO	0.534	−0.292	0.826
LiO	0.493	−0.729	1.222	BeF	0.558	−0.484	1.042	SiF	0.552	−0.229	0.781
NaC1	0.238	−0.809	1.048	BeCl	0.275	−0.475	0.750	AlO1	0.504	−0.445	0.949
NaC2	0.214	−0.814	1.028	BC1	0.227	−0.079	0.306	AlO2	0.513	−0.449	0.962
NaC3	0.190	−0.818	1.008	BC2	0.218	−0.088	0.306	AlF	0.592	−0.417	1.009
NaC4	0.246	−0.812	1.059	BC3	0.262	−0.140	0.402	GaF	0.575	−0.323	0.898
NaC5	0.258	−0.817	1.075		0.262	−0.063	0.325	CF	0.548	0.187	0.362
NaC6	0.271	−0.819	1.090	BC4	0.259	−0.087	0.346		0.548	0.210	0.339
NaN1	0.313	−0.756	1.068		0.259	−0.071	0.330				
NaO	0.480	−0.759	1.239	BN1	0.380	−0.093	0.474				
BeC1	0.229	−0.484	0.713	BN2	0.461	−0.092	0.554				
BeC3	0.244	−0.465	0.709	BN4	0.466	−0.085	0.551				

As obvious from these data, the complexes investigated in this work display a large range of hydrogen charges reflecting the breadth of the different DHB scenarios covered. Protonic H–atom charges range from 0.166 e in **BS1** and **BS2** to 0.592 e in **AlF** (see Table 1) and hydridic H–atom charges from −0.061 e in **AB2** (see Table 2) to −0.819 e in **NaC6** (see Table 1). It is noteworthy that some of the hydridic charges in transition metal and aromatic compounds are positive, which will be further elucidated below. Figure 3b shows the correlation between Δq and the DHB bond strength as reflected by $k^{a}$(DHB). As obvious from the large scattering of data points, there is no direct correlation between these two quantities, revealing that differences in the atomic charges is only one of the components determining the strength of these complex interaction. Therefore, Δq, although convenient, cannot serve as a direct DHB strength measure.

**Table 2 molecules-28-00263-t002:** Protonic and hydridic NBO charges (in e) for Group II–V. Protonic hydrogen charge q(H${}^{\delta +}$); hydridic hydrogen charge q(H${}^{\delta -}$); charge difference Δq = q(H${}^{\delta +}$) − q(H${}^{\delta -}$).

Group	Complex	q(H${}^{\delta +}$)	q(H${}^{\delta -}$)	Δq	Group	Complex	q(H${}^{\delta +}$)	q(H${}^{\delta -}$)	Δq
Aminoboranes	AB1	0.448	−0.065	0.512	Transition	FeN1	0.370	0.106	0.264
		0.448	−0.065	0.512	metals	FeN2	0.350	0.132	0.218
	AB2	0.445	−0.061	0.506		FeO1	0.483	0.074	0.409
		0.445	−0.064	0.509		FeO2	0.490	0.133	0.357
		0.441(H1)	(H2) − 0.052	0.493		FeF1	0.567	0.047	0.519
		0.441(H1)	(H3) − 0.062	0.503		FeF2	0.592	0.130	0.462
	AB3	0.443	−0.058	0.502	Noble gases	KrO	0.464	−0.300	0.765
		0.435(H1)	(H2,H3) − 0.056	0.491		BeAr	0.270	−0.510	0.781
	AB4	0.296	−0.063	0.359		BeKr	0.147	−0.502	0.649
		0.296	−0.063	0.359	Aromatic	Ar1	0.204	0.204	0.000
	AB5	0.494	−0.074	0.568	compounds	Ar2	0.204	0.193	0.011
	AB6	0.388	−0.065	0.453		Ar3	0.202	0.202	0.000
	AB7	0.571	−0.088	0.659					

*DHB binding energies, bond lengths and bond angles and local mode force constants:*Figure 4 shows how BEs and geometrical features that are frequently used to characterize the strength of a DHB, correlate with the local mode force constants $k^{a}$(DHB). Because BE includes geometry relaxation and electron density reorganization effects of the fragments upon DHB dissociation, as discussed above, there is no significant correlation between BE and $k^{a}$(DHB) as expected (R${}^{2}$ = 0.6186; see Figure 4a), in particular not for the broad range of complexes investigated in this work, leading to a variety of different fragments formed during the breakage of the DHB and disassociation into fragments. Nevertheless, we observed some trends: e.g., larger BEs are associated with larger $k^{a}$(DHB) values, reflecting stronger DHBs for the **BeO** complexes, whereas weak DHBs are characterized by small BE and $k^{a}$(DHB) values, such as **LiC1** or **LiC2**.

Figure 4b shows an overall power relationship between DHB lengths R(DHB) and $k^{a}$(DHB) following the generalized Badger rule of Cremer and co–workers [68]. Again, no significant correlation (R${}^{2}$ = 0.7014) is observed despite some trends; one of the strongest DHBs found in this study, namely for complexes **BeO1**, **BeO2**, and **BeO3** are characterized by smaller R(DHB) values (1.237, 1.170 and 1.091 Å, respectively), whereas the weakest DHBs found e.g., for **LiC1** and **BC4** have the longer DHBs with values of 2.495 Å and 2.533 Å, respectively. However, there are also outliers, such as **FeF2** which also has one of the weakest DHBs but a DHB length of 1.802 Å. Also obvious is that the Group V members do not follow this generalized Badger type rule caused by the specific topology of the bay H atoms forming these interactions. A more detailed discussion if these interaction can be considered as DHBs, can be found below.

The overall shortest DHB within Group I–IV is R(DHB) 1.091 Å (**BeO3**), which is still 0.348 Å longer than the R(HH) distance of 0.743 Å of the hydrogen molecule **H2**. Therefore, in our set of complexes we did not find any realization of a X${}^{\delta +}\cdots$H–H⋯Y${}^{\delta -}$ situation, suggested by Bakhmutov [11].

Figure 4c,d check the hypothesis that the angle X–H${}^{\delta +}\cdots$H${}^{\delta -}$ of the protonic fragment is closer to linear, whereas the hydridic fragment H${}^{\delta +}\cdots$H${}^{\delta -}$–Y is more bent and if there is a correlation between these angles and the DHB bond strength. We do not find any such correlation reflected by the large scattering of data points in Figure 4c,d. Furthermore, we also cannot confirm that the protonic part of the complex is more linear, whereas the hydridic part is more bent.

*Local mode versus normal mode frequency red/blue shifts:* Frequently normal mode frequency red/blue shifts are utilized to characterize DHB and HB. Although the frequency shifts of the DHB or HB normal mode stretching frequencies can be used to detect these bonds [43,69,70,71], they are not necessarily suited to assess the bond strength of these interactions as we discussed for HBs [67]. The normal mode frequency shift Δω=ω(complex)−ω(monomer) can only be related to the thermochemical strength of the HB or DHB in question, if the HB or DHB normal stretching modes in both complex and monomer are localized, which is generally not the case [67,72]. Therefore, we draw upon to local mode frequency shifts instead.

**Figure 4 molecules-28-00263-f004:**
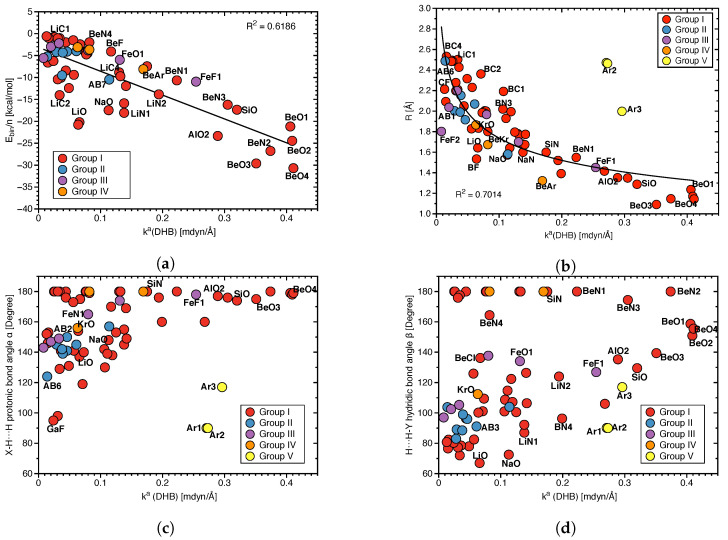
(**a**) Correlation between binding energy E${}_{bin}$ scaled by the number of DHBs *n* and $k^{a}$(DHB), (**b**) correlation between distance R(DHB) and $k^{a}$(DHB), (**c**) correlation between protonic bond angle (X–H${}^{\delta +}\cdots$H${}^{\delta -}$) and $k^{a}$(DHB), (**d**) correlation between hydridic bond angle (H${}^{\delta +}\cdots$H${}^{\delta -}$–Y) and $k^{a}$(DHB).

Figure 5a depicts the local mode frequency shift Δω of the protonic parts of DHB and Figure 5b that of the hydridic parts presented as a bar diagram. For both the DHB donor and acceptor, we found almost exclusively red shifts, i.e., the weakening of the bond in question upon complexation; however, the magnitude of the shifts differ significantly; protonic parts of DHBs shifts range from 139 to −2421 cm${}^{-1}$, whereas hydridic parts shifts cover a considerably smaller range, namely from 18 to −406 cm${}^{-1}$. It is interesting to note that largest red shifts reflecting large changes in DHB donor and acceptor upon DHB formation occur for **BeO3** and **BeO4**, i.e., the two complexes with the strongest DHBs.

In the following, we focus on evaluating to what extent the normal vibrational modes are localized in the DHB donor and/or acceptor of these complexes, which forms the necessary prerequisite for a red/blue shift discussion based on normal vibrational modes. As assessment tool we used in this work CMN to probe the local character of both protonic and hydridic normal stretching modes in complex and references molecules. As an example the CMN for the complex **BeO3** and its two references **HOCL2** and **BeH** is shown in Figure 6a,b.

The normal vibrational stretching mode of the DHB protonic part of reference **HOCL2** (O1H4, yellow bar in Figure 6a) with a frequency of 3589 cm${}^{-1}$ is 100% localized. In contrast, the hydridic normal vibrational stretching modes of reference **BeH** with frequencies of 2203 and 1995 cm${}^{-1}$ are composed of 50% Be2H3 (red color) and 50% Be2H1 (orange color), reflecting the symmetry of the **BeH** reference molecule. Upon complex formation, the protonic part of DHB is considerably weakened, and most importantly, as reflected in Figure 6a, it strongly couples with other modes in the **BeO3** complex. In **BeO3**, the normal mode corresponding to the complex normal vibration of 1386 cm${}^{-1}$ has 47% O5H4 (the protonic part of DHB, yellow color) and 39% H3H4 (DHB, purple color); see Figure 6a. The normal vibrational modes corresponding to 2286 cm${}^{-1}$ and 2101 cm${}^{-1}$ have 14% and 58% B3H3 (the hydridic bond) character. This clearly shows (i) the strong influence of the protonic part of DHB on DHB and (ii) that a red/blue shift discussion based on normal vibrational mode frequencies is rather questionable, and one has to resort to local mode frequencies for a meaningful analysis, which is further elucidated in the following.

**Figure 5 molecules-28-00263-f005:**
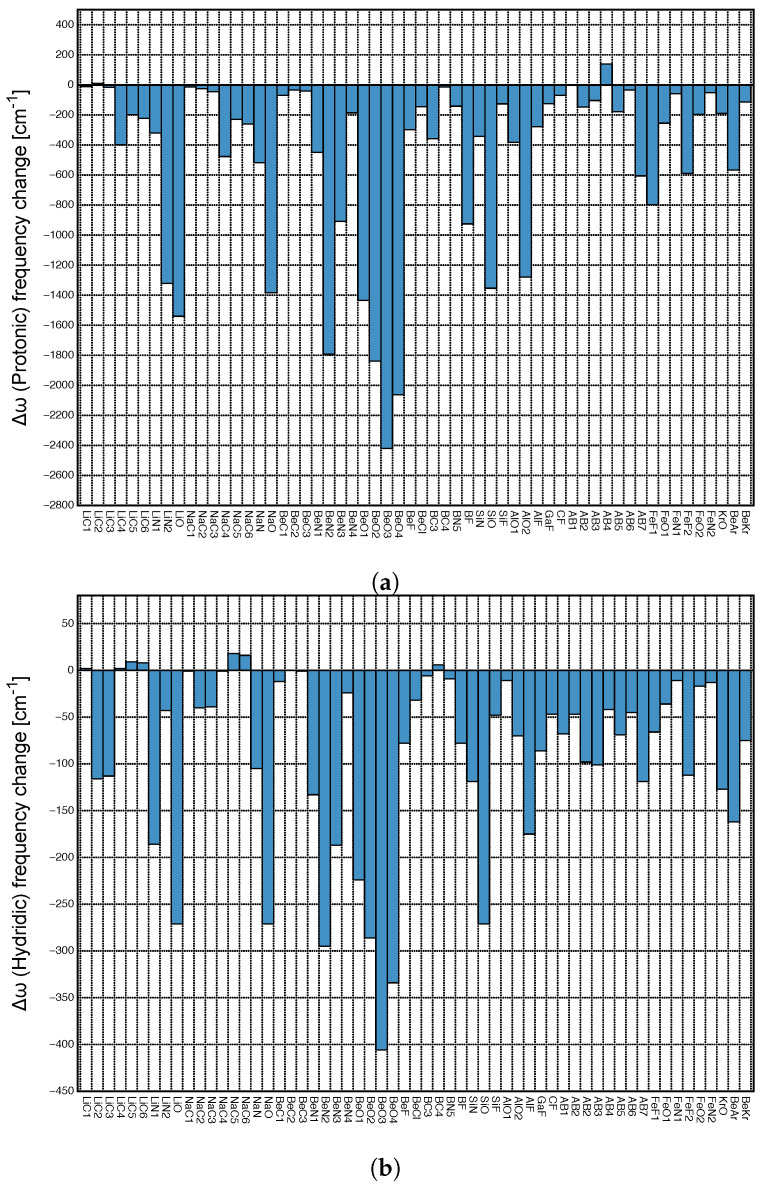
(**a**) Local mode frequency shifts Δω of the protonic parts of DHBs. (**b**) Local mode frequency shifts Δω of the hydridic parts of DHBs, where a positive sign denotes a blue–shift and a negative sign a red–shift.

Figure 7a presents the relationship between protonic and hydridic local mode frequency shifts Δω, and Figure 7b the relationship between the corresponding relative shifts Δω/ω, involving Group I–Group IV members.

**Figure 6 molecules-28-00263-f006:**
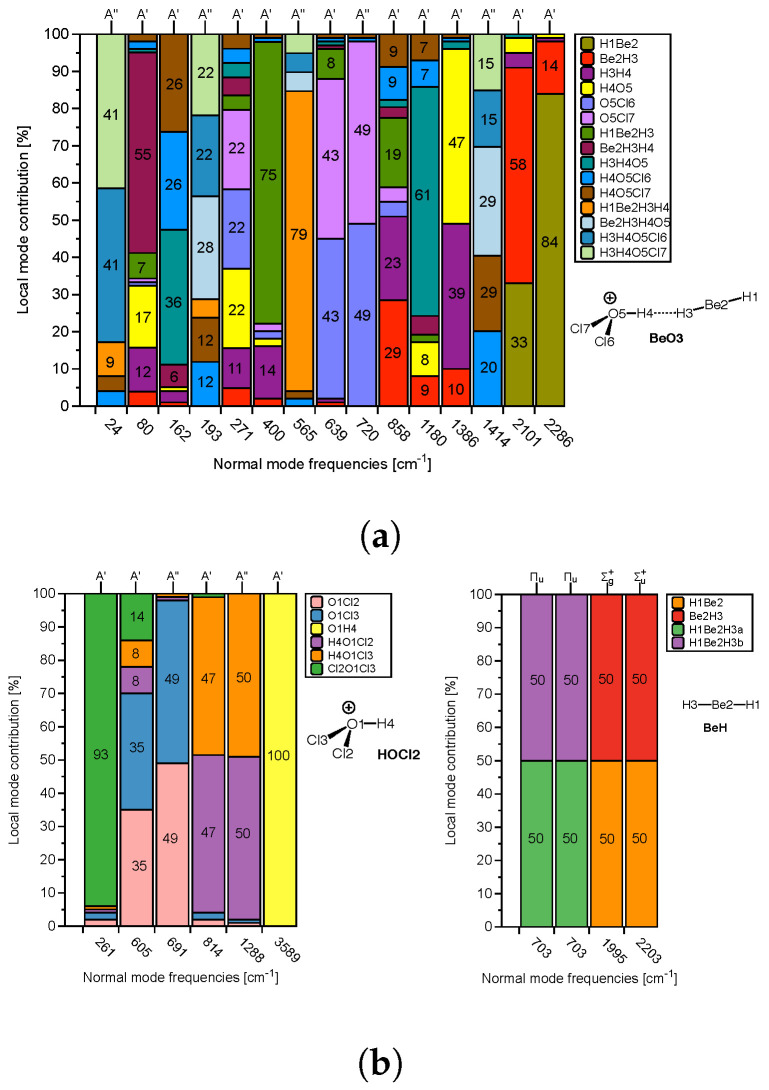
(**a**) CNM analysis for the **BeO3** complex. (**b**) CNM analysis for the two reference molecules **HOCL2** (DHB donor) and **BeH** (DHB acceptor) forming the complex.

**Figure 7 molecules-28-00263-f007:**
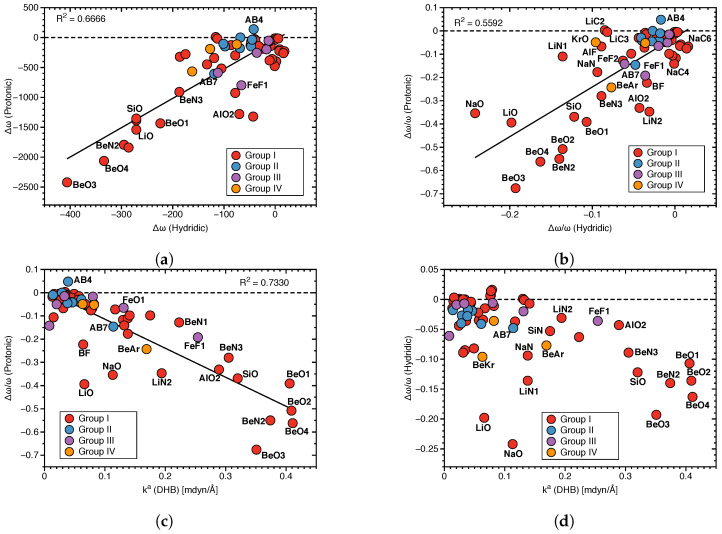
(**a**) Correlation between local mode frequency shifts Δω(Hydridic) and Δω(Protonic). (**b**) Correlation between the relative frequency shifts Δω/ω(Hydridic) and Δω/ω(Protonic). (**c**) Correlation between Δω/ω(Protonic) and $k^{a}$(DHB). (**d**) Correlation between Δω/ω(Hydridic) and $k^{a}$(DHB).

There is no significant correlation between hydridic and protonic frequency shifts (R${}^{2}$ = 0.6666 and 0.5592, respectively; see Figure 7a,b), although some trends are visible. Overall, larger protonic frequency shifts are in line with larger hydridic frequency shifts, as found for **BeO3**, **BeO4** and on the other end of the spectrum smaller protonic frequency shifts are in line with smaller hydridic frequency shifts, as found for **AB4**. However, there are also a number of outliers, such as **LiO** and **NaO** complexes. Because of the large difference in the frequency shift values for protonic and hydridic parts of DHBs, for the following correlations only the latter were used.

Next, we explored the question if there is a correlation between the frequency shifts of the protonic and hydridic parts of DHB, and the strength of the DHB, as suggested in the literature [43,69,70]. Figure 7c shows the correlation between the relative frequency shifts Δω/ω(Protonic) and the local mode force constant k${}^{a}$(DHB), and Figure 7d shows the same situation for Δω/ω(Hydridic). We found some general trends for the protonic parts of DHBs shifts (R${}^{2}$ = 0.7330). Stronger DHBs are characterized by larger red shifts, as found for **BeO2**, **BeO4** and weaker DBH are characterized by small red shifts or even small blue shifts, such as in the case of **AB4**. Although the same overall trend can be observed for the hydridic frequency shifts scattering of data points, and the outliers **LiO** and **NaO** are more pronounced. This shows that the strength of the DHB cannot be directly deduced from the frequency shifts of the protonic or hydridic parts of DHBs upon complex formation.

Figure 8a shows the correlation between protonic frequency shifts and binding energies and Figure 8b the corresponding correlation for the hydridic frequency shifts. Again, in both cases no significant correlation could be found (R${}^{2}$ = 0.7768 and 0.5933, respectively; see Figure 8a,b), except the general trend that larger binding energies corresponding to stronger DHBs are characterized by larger red shifts, whereas smaller binding energies corresponding to weaker DHBs are characterized by smaller red shifts or even a small blue shift. These findings are consistent with the observed correlation between frequency shifts and local mode DHB force constants. Figure 8c,d show the correlation between frequency shifts and scaled energy density H${}_{c}$/$\rho_{c}$ for the protonic and hydridic parts of DHB, respectively. In both cases, there is no correlation between these two quantities in contrast to $H_{c}/\rho_{c}$(DHB), which shows at least a weak correlation with k${}^{a}$(DHB) (see Figure 3a).

**Figure 8 molecules-28-00263-f008:**
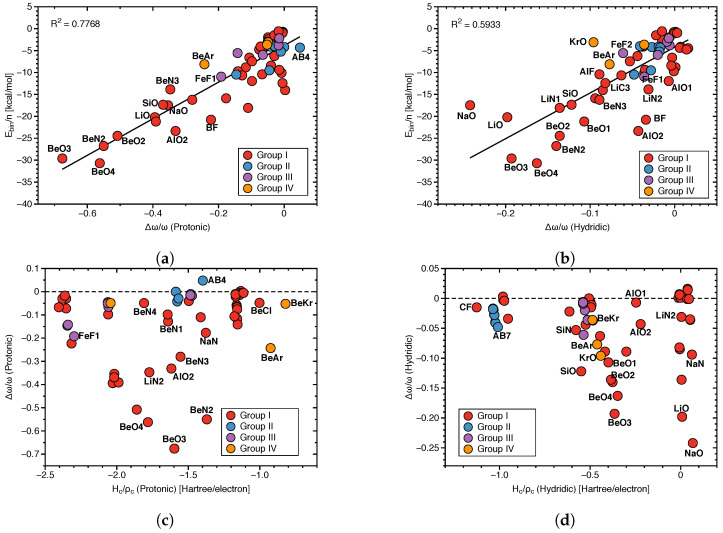
(**a**) Correlation between E${}_{bin}$/n and Δω/ω(Protonic). (**b**) Correlation between E${}_{bin}$/n and Δω/ω(Hydridic). (**c**) Correlation between $H_{c}/\rho_{c}$(Protonic) and Δω/ω(Protonic). (**d**) Correlation of $H_{c}/\rho_{c}$(Hydridic) and Δω/ω(Hydridic).

### 2.2. Some Group Specific Highlights

*Group I:* This group represents DHBs being built from hydrides, such as LiH, NaH, BeH${}_{2}$, BH${}_{4}^{-}$, and AlH${}_{4}^{-}$ interacting with simple organic molecules—for example, $H_{2}O$, HNO, ${\mathrm{CH}}_{4}$, ${\mathrm{CFH}}_{3}$, and ${\mathrm{CF}}_{2}H_{2}$—and also some cations, such as $H_{3}O^{+}$, NH${}_{4}^{+}$, ${\mathrm{CFH}}_{4}^{+}$, taken from the work of Bakhmutov [11]. With 52 members, this is the largest group investigated in this work, with DHB lengths stretching from 1.091 Å to 2.533 Å and BSOs from 0.159 to 0.415. Among them, the strongest H−H interactions were observed for the molecular BeO clusters and the weakest interaction was observed for **CF** sharing three DHBs. It is interesting to note that **BC4** also sharing three DHBs has a longer DHB length of 2.533 Å, compared to 2.484 Å found for the DHBs of **CF**; however, both **CF** and **BC4** DHBs are of comparable strengths (BSO n = 0.159 versus BSO n = 0.168, respectively), another exemption of the generalized Badger rule.

As discussed above, we observed larger changes in the local mode frequencies of the protonic parts of DHB than for the hydridic parts upon DHB complex formation, a trend which also applies to Group I representatives. An interesting example is the **BeO3** complex that has a DHB binding energy of −29.61 kcal/mol. The local mode frequency of the protonic H–O bond has a value of 3584 cm${}^{-1}$ in the **HOCL2** monomer, while in the DHB complex this frequency has a value of 1163 cm${}^{-1}$, which corresponds to a large red–shift of −2421 cm${}^{-1}$. On the other hand, in the similar **BeF** DHB complex with a DHB binding energy of −4.08 kcal/mol, the red–shift of the protonic bond H–F is with a value of −298 cm${}^{-1}$ almost ten times smaller. This large difference in the red–shifts can be explained by the different electron density transfer from the hydridic Be–H bond into the antibonding σ* orbital of the protonic H–O and H–F bonds. According to our calculations, the population of the H–O σ* orbital in the **BeO3** complex has a value of 0.248 e, while the population of the H–F σ* orbital in the **BeF** has only a value of 0.019 e, which is more than ten times smaller. These results quantify the findings of a previous study, suggesting that the experimentally observed red–shifts in DHBs depend on the amount of electron density transfer into the antibonding σ* orbitals of the protonic bonds [43]. The larger this transfer is the weaker becomes the protonic part of DHB. This is also reflected in the series of *charge assisted* DHBs [11] **BeO1**, **BeO2**, and **BeO3**. Starting from $H_{3}O^{+}$ successive chlorination, i.e., increasing the electrophilicity of the protonic group leads to a considerable decrease of the DHB length (1.237, 1.170, and 1.091 Å, respectively), however the DHB strength remains fairly unchanged (BSO n values of 0.414, 0.414, and 0.397).

*Group II:* This group includes dimer (**AB1**), trimer (**AB2**), and tetramer (**AB3**) complexes of ammonia–borane $H_{3}B-{\mathrm{NH}}_{3}$ as well as some reference complexes (**AB4**–**AB7**). Ammonia borane polymers have been suggested as storage medium for hydrogen [73], therefore we assessed these systems how, starting from **AB1**, the strength of the H−H interaction changes. According to the results of our study the H−H interaction between the monomeric units increases as reflected by the BSO values (0.199, 0.226, and 0.244, for **AB1**, **AB2**, and **AB3**, respectively), whereas the DHB distances remain fairly constant (2.003, 1.916, and 2.069 Å, for **AB1**, **AB2**, and **AB3** respectively). These results suggest an increasing DHB strength trend in larger ammonia borane clusters and ammonia borane polymers. An interesting alternative is **AB7**, namely paring ammonia borane with HF leading to the shortest and strongest DHB in this group (BSO = 0.291 and R(DHB) = 1.582 Å). A follow–up study is planned.

*Group III:* Group 3 includes transition metal clusters made up of iron–phosphine complexes forming either one or two DHBs with ammonia, water, or HF. DHB length range from 1.451 to 2.198 Å wherein forming two DHB interactions (**FeN2**, **FeO2**, and **FeF2**) leads to longer and weaker DHB interactions than those for the complexes with one DHB (**FeN1**, **FeO1**, and **FeF1**) which parallels our observations for Group I members. Unsurprisingly, HF forms the shortest and strongest DHB interactions in this group (**FeF1**: R(DHB) = 1.451 Å, BSO n = 0.363). As compared to other groups, Group III shows smaller local mode red shifts for both protonic and hydridic parts of DHBs, with the smallest protonic and hydridic part shifts found for **FeN1** (−0.017 and −0.006 cm${}^{-1}$, respectively). Group III members overall show the best correlation between protonic and hydridic frequency shifts in relation to the other groups, likely because the shifts themselves are so small. We also observe a stronger correlation between frequency shifts and BEs than for any of the other groups. However, Group III members are also the most similar compared to the members in other groups, which may cause the better correlation.

The BEs of Group III range from −3.81 to −11.14 kcal/mol for **FeN1** and **FeF2**, respectively, which are among the smallest BEs of the entire set of complexes investigated in this work. Thus, we assert that such transition metal complexes do not tend to form strong DHBs. This difference from other groups can be explained by the presence of d–orbitals, which can withdraw electron density from antibonding σ* orbitals of the hydridic parts of DHBs, leading to complexes with predominately electrostatic character, as reflected by normalized energy density $H_{c}/\rho_{c}$ shown in Figure 3, with the exception of **FeF1** and **FeF2**. This is an interesting aspect to be further explored in the future.

*Group IV:* This group features four noble gas complexes, whose DHB bond lengths range from 1.322 to 1.866 Å, with BSO n values between 0.246 and 0.324. Although these DHB strengths are on the lower end of the spectrum, they are predominantly covalent in nature, as reflected by the normalized energy density $H_{c}/\rho_{c}$ values shown in Figure 3. For Group IV members, we find exclusively local mode red shifts indicating the weakening of both the protonic and hydridic part of DHB upon complexation. **BeAr** shows with −0.243 the most significant relative protonic shift in this group, as well as the largest BSO n value (0.324). This can be explained by the linearity of the DHB as well as the presence of fewer shielding electrons in argon. As with the other groups, there is a weak correlation between the frequency shifts and BE.

*Group V:* Naphthalene, anthracene and phenanthrene **Ar1**–**Ar3** have been added to the set of DHB complexes in order to compare the H−H interaction in these aromatic systems, where both hydrogen atoms involved are equally charged, with the DHB properties of the other groups investigated in this work. In previous work [66] we could disprove arguments in favor of DHB based on the existence of a bond path and bond critical point between these two H atoms and the stability of anthracene [11,22,65] as dubious [66]. The sole existence of a bond path and a bond critical point does not necessarily imply the existence of a chemical bond or weak chemical interaction, it may be just an artifact of a closer contact between the two atoms in question [66,74,75,76,77,78,79,80]. In addition, previous attempts to explain the higher stability of phenanthrene via a maximum electron density path between the bay H atoms are misleading in view of the properties of the electron density distribution in the bay region. The 6.8 kcal/mol larger stability of phenanthrene relative to anthracene predominantly (84%) results from its higher resonance energy, which is a direct consequence of the topology of ring annulation [66]. In summary, these observations clearly disqualify the H−H in these systems as DHBs. With the higher level of theory used in this study, we could no longer find a bond path between the bay H atoms for **Ar1** and **Ar2**. The bond path and bond critical point found for **Ar3** are an artifact of the closed contact (R(HH) = 1.998 Å). The relative large BSO n values of 0.370, 0.370 and 0.379 (see Figure 2a) are also an artifact of the topology of the bay H atoms. In addition, Figure 2d,f as well as Figure 3 and Figure 4b,c clear identify these complexes as outliers.

*Group VI:* This is a special group devoted to modeling of the strength of DHB in hydrogen clusters, which have been suggested in the interstellar space [81]. Figure 9a–c present results of our investigations for molecular clusters **H4P**, **H5P**, **H6P**, **H7P**, and **H9P** involving diatomic hydrogen **H2**, and triatomic hydrogen cation **H3P**. The BSO data as a function of the local mode force constants k${}^{a}$ of DHBs for these clusters are shown in Figure 9a, whose values are based on the reference molecules presented in the computational details of this study. According to Figure 9a, the strongest DHB (k${}^{a}$ = 5.835 mdyn/Å) is for diatomic hydrogen **H2**, and the weakest (k${}^{a}$ = 0.101 mdyn/Å) is for DHB between the H${}_{3}^{+}$ and H${}_{2}$ units of **H7P**. The strengths of DHB for this Group investigated in our study correlate (R${}^{2}$ = 0.9218) with the DHB lengths, as shown in Figure 9b. The shortest DHB (R = 0.743 Å) is for **H2**, and the longest (R = 1.675 Å) for DHB between the H${}_{3}^{+}$ and H${}_{2}$ units of **H9P**. Similarly, according to Figure 9c there is a correlation observed in our study between the strengths of DHB and the covalent characters of DHB expressed by the normalized energy densities H${}_{c}$/$\rho_{c}$, although this correlation is less perfect (R${}^{2}$ = 0.8981). The strongest DHB of **H2** has the most negative normalized energy density (H${}_{c}$/$\rho_{c}$ = −1.125 Hartree/electron), while the weakest DHB between the H${}_{3}^{+}$ and H${}_{2}$ units of **H7P** has one of the less negative normalized energy density (H${}_{c}$/$\rho_{c}$ = −0.194 Hartree/electron). Generally, we observed in our investigation of this Group, formation of weak DHBs between the H${}_{3}^{+}$ and H${}_{2}$ units (light blue and orange colors in **H7P** and **H9P** in Figure 9), whose strength is smaller than the strength of DHBs within the H${}_{3}^{+}$ unit (dark blue, red, and brown colors in **H7P** and **H9P** in Figure 9), and the strength of DHB in the H${}_{2}$ unit of these clusters (violet color in **H7P** and **H9P** in Figure 9). The **H6P** cluster represents formation of weak DHBs (yellow color in **H6P** in Figure 9) between the H${}_{2}^{+}$ and H${}_{2}$ units (dark blue color in **H6P** in Figure 9), while the **H5P** cluster shows formation of weak DHBs between the H${}^{+}$ and H${}_{2}$ units (red color in **H5P** in Figure 9). However the **H4P** cluster presents formation of weak DHB between the H${}_{3}^{+}$ unit and the H atom (orange color in **H4P** in Figure 9). Generally, according to our calculations, the hydrogen clusters are formed between the positively charged unit of the cluster (H${}^{+}$ in **H5P**, H${}_{2}^{+}$ in **H6P**, and H${}_{3}^{+}$ in **H7P** and **H9P**) and the H${}_{2}$ unit. Therefore, we conclude that the results of this study are suggesting possible applications for hydrogen storage devices, where a positively charged central kernel of a molecular complex is surrounded by H${}_{2}$ molecules, which can easily dissociate.

**Figure 9 molecules-28-00263-f009:**
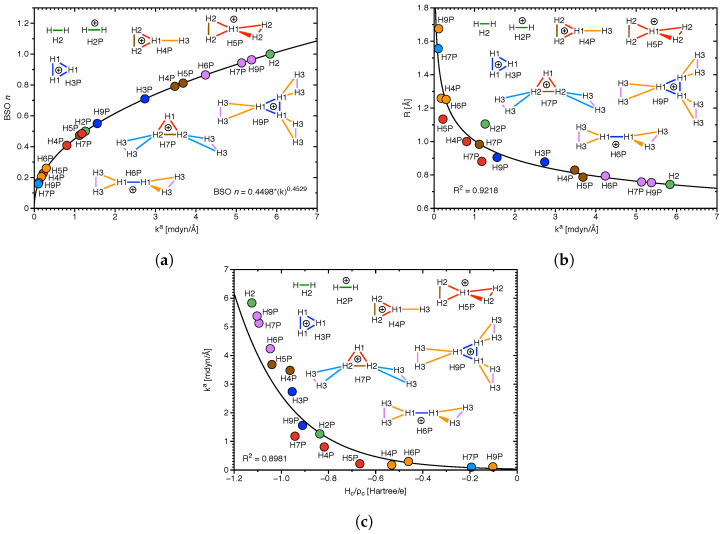
(**a**) BSO n(HH) as a function of the local mode force constant k${}^{a}$(HH); power relationship based on H${}_{2}$ and H${}_{2}^{+}$ references see text. (**b**) Correlation of distance R(HH) and k${}^{a}$(HH). (**c**) Correlation of k${}^{a}$(HH) and normalized energy density $H_{c}/\rho_{c}$.

**Table 3 molecules-28-00263-t003:** NBO charges (e) for the hydrogen clusters.

	Complex	Atom	q	Bond	$\Delta q^{a}$
Group VI	H2		0.000	H-H	0.000
	H2P		0.500	H-H	0.000
	H3P		0.333	H1-H1	0.000
	H4P	H1	0.161	H1-H2	0.010
		H2	0.171	H2-H2	0.000
		H3	0.496	H3-H1	0.334
	H5P	H1	0.281	H1-H2	0.102
		H2	0.180	H2-H2	0.000
	H6P	H1	0.219	H1-H1	0.000
		H3	0.141	H1-H3	0.078
	H7P	H1	0.209	H1-H2	0.075
		H2	0.284	H2-H2	0.000
		H3	0.056	H2-H3	0.228
				H3-H3	0.000
	H9P	H1	0.262	H1-H3	0.227
		H3	0.035	H3-H3	0.000

## 3. Methods and Computational Details

As a major assessment tool of H–H bond strength, we utilized LMA in this work. A comprehensive review of the underlying theory and recent applications can be found in Refs. [72,82]. Therefore, in the following, only some essentials are summarized.

Normal vibrational modes are generally delocalized as a result of kinematic and electronic coupling [83,84,85]. A particular vibrational stretching mode between two atoms of interest can couple with other vibrational modes such as bending or torsion, which inhibits the direct correlation between normal stretching frequency or associated normal mode force constant and bond strength, as well as the comparison between normal stretching vibrational modes in related molecules. As a consequence, the normal stretching force constant cannot be used as a direct bond strength measure. One needs to derive local counterparts that are free from any mode–mode coupling. Konkoli, Cremer and co–workers solved this problem by solving the mass–decoupled analogue of Wilson’s equation of vibrational spectroscopy [53,54,55,56,57], leading to local vibrational modes, associated local mode frequencies and local mode force constants.

There is a 1:1 relationship between the normal vibrational modes and a complete non–redundant set of local vibrational modes via an *adiabatic connection scheme* (ACS) [86], which forms the foundation for a new method of analyzing vibrational spectra, the characterization of normal mode (CNM) procedure [55,56,57,66,72,86,87,88,89,90,91]. CNM decomposes each normal mode into local mode contributions leading to a wealth of information about structure and bonding. Recently, CNM has been successfully applied in the investigation of pK${}_{a}$ probes [92] and to assess the qualification of *vibrational Stark effect* probes [93].

Zou and Cremer showed that the local stretching force constant $k^{a}$ reflects the curvature of the PES in the direction of the bond stretching [94]. This important result qualifies the local stretching force constants $k^{a}$ as unique quantitative measures of the intrinsic strength of chemical bonds and/or weak chemical interactions based on vibrational spectroscopy. Therefore, LMA has been extensively applied for the study of chemical bonding and non–bonded interactions, as documented in Refs. [72,82] and references cited therein, including the characterization of HB ranging from conventional HBs [67], unconventional HBs [95,96], HBs in water and ice [87,97,98,99,100,101,102], and HBs in biomolecules [103,104,105,106] to HBs in catalysis [107,108]—a list which has been extended by DHB in this work. The local vibrational modes theory allows us to analyze a variety of internal coordinates in addition to bond lengths, such as angles between bonds, dihedral bond angles, puckering coordinates, and others. However, this study is focused on the bond lengths leading to the local mode force constants directly related to the strength of DHB.

The widely accepted definition of a DHB includes the following characteristics: DHBs H${}^{\delta +}\cdots$H${}^{\delta -}$ are formed from two H–atoms with opposite partial charges and a distance R(DHB) between the two H–atoms that is smaller than the sum of their van der Waals radii (2.4 Å); the angle X–H${}^{\delta +}\cdots$H${}^{\delta -}$ of the protonic fragment is closer to linear, whereas the hydridic fragment H${}^{\delta +}\cdots$H${}^{\delta -}$–Y is more bent [11]. For a larger separation, it has been suggested to apply a criterion based on the topological electron density: The density $\rho_{c}$ at the bond critical point $r_{c}$ between the two H–atoms should be small, and the Laplacian $\nabla^{2}$($\rho_{c}$) should be small and positive. Whereas bond energies are assumed to be similar for both HBs and DHBs, DHBs should have a smaller covalent character, i.e., a more pronounced electrostatic nature [11]. As discussed above, in addition to R(DHB) [1,11], the red/blue shifts of the normal mode frequency associated with the DHB [9,43], as well as components of the binding energies (BE) determined with different energy decomposition methods [40,41,42] have been applied as a measure of the DHB bond strength.

However a caveat is necessary: both the bond dissociation energies (BDE) and BE [109,110,111] as well as associated energy decomposition schemes (EDA) [112,113,114], bond distances [115,116] and normal mode frequency shifts, are not suited to measure direct bond strength. The BDE is a reaction parameter that includes all changes taking place during the dissociation process. Accordingly, it includes any (de)stabilization effects of the fragments to be formed. It reflects the energy needed for bond breaking, but also contains energy contributions due to geometry relaxation and electron density reorganization in the dissociation fragments [67]. Therefore, the BDE is not a suitable measure of the intrinsic strength of a chemical bond and its use may lead to misjudgments, as documented in the literature [117,118,119,120,121,122]. In addition, EDA schemes are not free from arbitrariness [123]. Additionally, the bond length is not always a qualified bond strength descriptor. Numerous cases have been reported, illustrating that a shorter bond is not always a stronger bond [54,55,57,124,125].

The use of normal mode stretching frequencies and/or frequency shifts as convenient bond strength measures (both are accessible experimentally and computationally) is problematic, because of the coupling of the stretching mode with other vibrational modes, which hampers a direct comparison between bonds in strongly different systems, as pointed out above. The topological analysis of the electron density can be useful to uncover possible attractive contacts between two atoms via the existence of a maximum electron density path (i.e., bond path), with a bond critical point connecting the two nuclei under consideration [126]. However, the sole existence of a bond path and a bond critical point does not necessarily imply the existence of a chemical bond; in particular, the QTAIM description of weak chemical interactions may be problematic [46,66,74,75,76,78,79]. Using the Laplacian $\nabla^{2}$($\rho_{c}$) as a complementary measure is problematic too [127,128,129], because it does not reflect the complex interplay between kinetic and potential energy, which accompanies the bond forming process [130,131].

Therefore, in this work we utilized LMA to derive a quantitative DHB bond strength measure, in particular local mode force constants and related bond strength orders. It is convenient to base the comparison of the bond strength for a set of molecules on a chemically more prevalent bond strength order (BSO n) rather than on a comparison of local force constant values. Both are connected via a power relationship according to the generalized Badger rule derived by Cremer and co–workers [68]: BSO n = A ($k^{a}$)${}^{B}$. Constants A and B can be determined from two reference compounds with known BSO n values and the requirement that, for a zero force constant, the BSO n must be zero. For the protonic parts of DHBs, the hydridic parts of DHBs, and the H–H interaction in DHBs investigated in this study, we use as references the FH bond in the FH molecule with BSO n = 1 and the FH bond in the [F⋯H⋯F]${}^{-}$ anion with BSO n = 0.5 [67,99,103,106]. For the ωB97X-D [132,133] /aug-cc-pVTZ [134,135] model chemistry, applied in this study, this led to $k^{a}$(FH) = 9.719 mdyn/Å, $k^{a}$(F⋯H) = 0.803 mdyn/Å and A=0.531 and B=0.278. For the DHBs in hydrogen clusters of Group VI investigated in this work, we used as references $H_{2}$ (BSO n = 1, $k^{a}$ = 5.840 mdyn/Å) and $H_{2}^{+}$ (BSO n = 0.5, $k^{a}$ = 1.263 mdyn/Å), leading to A=0.450 and B=0.453.

LMA was complemented with QTAIM [61,62,78,126], where in this work we applied the Cremer–Kraka criterion [127,128,129] of covalent bonding to assess the covalent/electrostatic character of DHBs. The Cremer–Kraka criterion is composed of two conditions: (i) the necessary condition—existence of a bond path and bond critical point $r_{c}$ = *c* between the two atoms under consideration; (ii) the sufficient condition—the energy density $H(r_{c})$ = $H_{c}$ is smaller than zero. H(r) is defined as H(**r)** = G(**r)** + V(**r)**, where G(r) is the kinetic energy density and V(r) is the potential energy density. A negative V(r) corresponds to a stabilizing accumulation of density, whereas the positive G(r) corresponds to the depletion of electron density [128]. As a result, the sign of $H_{c}$ indicates which term is dominant [129]. If $H_{c}<0$, the interaction is considered covalent in nature, whereas $H_{c}>0$ is indicative of electrostatic interactions. In addition to the QTAIM analysis, we calculated Natural Bond Orbital (NBO) [58,59] atomic charges and the corresponding charge transfer between the two hydrogen atoms involved in DHB, and we introduced local mode frequency shifts [67] as a more reliable parameter to distinguish different DHB situation than normal mode frequency shifts, which are contaminated by mode–mode coupling.

The geometries of all group members and references compounds (see Figure 1) were optimized using the ωB97X–D density functional [132,133] combined with Dunning’s aug–cc–pVTZ basis set [134,135]. The ωB97X–D functional was chosen as it has been proven to reliably describe weak (long–range) intermolecular interactions covering the diverse range of molecules considered in particular in combination with the augmented basis set [133,136]. Using the same model chemistry as applied in our comprehensive study on HBs [137] also allows the direct comparison of HB and DHB features. Furthermore, this model chemistry has been recently assessed and characterized as useful for the description of DHB [39]. Geometry optimizations were followed by a normal mode analysis and LMA. Geometry optimizations, normal mode analyses and NBO analyses were performed with the Gaussian09 program package [138]. For the subsequent LMA the program LModeA [139] was utilized. The QTAIM analysis was performed with the AIMAll package [140].

## 4. Conclusions

In this computational study, we investigated the strength and nature of dihydrogen bonding (DHB) in a variety of molecular complexes and clusters, categorized into six different groups. As an assessment tool, we used the Local Mode Analysis (LMA) based on DFT calculations involving dispersion corrections and a flexible basis set. LMA, which was used to determine the precise strength of individual DHBs in these complexes and clusters, was supported by the topological analysis of the electron density, quantifying the covalent versus electrostatic character of these DHBs. Along with the analysis of DHB, we also investigated the strength and the nature of the protonic bonds X–H${}^{\delta +}$, as well as the hydridic bonds H${}^{\delta -}$–Y forming the DHBs (i.e., X–H${}^{\delta +}\cdots$H${}^{\delta -}$–Y) for Group I–V members. According to our results, the strength of DHB is modulated by the strength of the protonic bonds X–H${}^{\delta +}$ rather than by the strength of the hydridic bonds H${}^{\delta -}$–Y, which was also confirmed for Group VI members involving hydrogen clusters. In accordance with our calculations, the hydrogen clusters were formed by weak DHB between the central positively charged H${}_{3}^{+}$ kernel of the cluster and the surrounding H${}_{2}$ molecules, which can easily dissociate, suggesting a possible application for hydrogen storage.

We hope that our study inspires the community to add the Local Mode Analysis to their repertoire for the investigation of chemical bonds and intermolecular interactions.

## Figures and Tables

**Figure 1 molecules-28-00263-f001:**
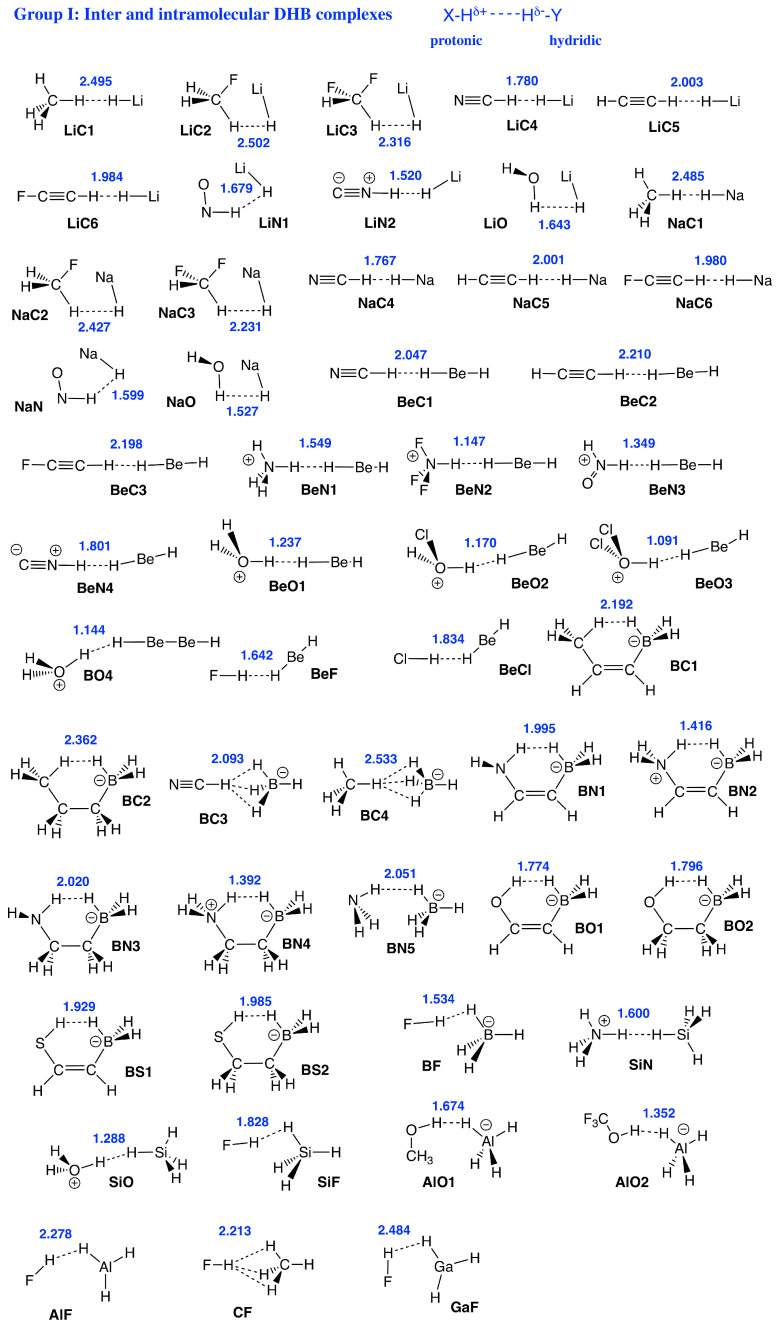
Sketches of Group I–Group VI members and reference compounds investigated in this study. DHB distances (in Å) are given in blue. Short names of the complexes used throughout the manuscript are given in black.

## Data Availability

All data supporting the results of this work are presented in tables and figure of the manuscript and in the Appendix A.

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
