# Peer review of "Dihydrogen Bonding—Seen through the Eyes of Vibrational Spectroscopy"

_molecules, 2022, doi:10.3390/molecules28010263_

Round 1

Reviewer 1 Report

The paper is very interesting and sound. I recommend some minor revisions:

- Use a single affiliation for all authors, with multiple e-mail addresses.

- the authors describe always the vibration modes as "stretching". What about bending or other complex vibrations?

- Figure 1: give a short description of the different groups when you introduce the figure in the text

- lines 169-174 are results of the study. They should be moved to the results section and removed from "methods".

- Also Figure 2 should be moved to results section

- in all captions, avoid the sentence "wB97X-D/aug-cc-pVTZ level of theory" an use some longer expression to indicate the level of theory and the basis set used for calculations or you can avoid completely this sentence as it is the only model used.

- line 256: remove "both"

Author Response

Use a single affiliation for all authors, with multiple e-mail addresses.

Done

The authors describe always the vibration modes as "stretching". What about bending or other complex vibrations?

The local vibrational modes theory allows to analyze a variety of coordinates, in addition to bond lengths, bond and dihedral angles can be analyzed as well as puckering coordinates, or other coordinates as long as the corresponding Wilson B-matrix can be derived. However, this study is focused on bond lengths and their local mode force constants directly related to the strength of DHB. We included a clarifying sentence in the Methods and Computational Details section of the corrected manuscript.

Figure 1: give a short description of the different groups when you introduce the figure in the text.

A more detailed description of the different groups has been added at the end of the introduction.

Lines 169-174 are results of the study. They should be moved to the results section and removed from "methods.

Lines 169-174 describe the parameters of the power relation between BSO and local mode force constant, which we used in our study to obtain the BSO values shown in the Results section. Therefore, we prefer to keep this part in the Methods and Computational Details section rather than moving it to the Results section.

This is in line with previously published work.

Also Figure 2 should be moved to results section.

Done

In all captions, avoid the sentence "wB97X-D/aug-cc-pVTZ level of theory" an use some longer expression to indicate the level of theory and the basis set used for calculations or you can avoid completely this sentence as it is the only model used.

The "wB97X-D/aug-cc-pVTZ level of theory" sentence has been removed from the figures and tables captions in the revised version of the manuscript.

Line 256: remove "both”.

Done

Reviewer 2 Report

The manuscript written by M. Freindorf and coworkers reports an interesting study on the nature ad features of the so-called dihydrogen bonding (DHB). To conduct their investigations, the authors took advantage of the Local Vibrational Mode Analysis (LMA), a technique that allows to obtain local vibrational modes along with the corresponding local mode frequencies and local mode force constants, which are quantities directly associable with the force of single bonds. In particular, using LMA and other computational strategies (notably, QTAIM and NBO population analysis), Freindorf and collaborators explored the correlations between DHB strengths (in the “form” of local mode force constants) and quantities/descriptors that are usually considered to evaluate the force of dihydrogen bonds, such as binding energies, bond distances and frequency red/blue shifts. Interestingly, in some situations the obtained results overturned the common belief or correlations were not as strong as one should expect.

Overall, I believe that the paper should be considered as publishable I Molecules after the authors will have addressed the following minor points:

1)     I found a bit the terms “protonic hydrogen bond” (protonic HB) and “hydridic hydrogen bond” (hydridic HB) misleading. If I understood correctly, these are not hydrogen bonds (in the sense of non-covalent interactions), but bonds that involve hydrogen atoms. In the studied systems, the only non-covalent interactions are the di-hydrogen bonds. I strongly suggest the authors to modify the terms throughout the text as follows (for example): “protonic H-E bond” and “hydridic H-E bond” (where E stands for element). Of course, the authors can propose other alternatives, but not “protonic hydrogen bond” (protonic HB) and “hydridic hydrogen bond” (hydridic HB).

2)     Table 3 is not discussed at all in the main text. If the authors want to keep it, they should move it to the Supplementary Material.

3)     Page 9 – Lines 260-261. The authors wrote: “It is noteworthy that some of the hydridic changes in transition metal and noble gas compounds are positive”. By inspecting Table 2, the hydridic charges in noble gas compounds are always negative. On the contrary, those of the aromatic compounds are always positive. The authors should correct this point.

4)     Page 10 – Lines 297-298. The authors wrote: “Furthermore, we also cannot confirm that the protonic part of the complex is more bent whereas the hydridic part is more linear”. Is it not the opposite considering what was discussed few lines above? The authors should correct this sentence or be clearer on this point.

5)     Figure 5. This figure must be significantly improved. By printing the manuscript, the numbers or the names of the complexes/components are not readable at all (too small). The authors should absolutely re-edit the figure.

6)     Page 15 – Lines 408-410. The sentence “Ammonia borane… interaction changes” is really unclear and must be rephrased.

7)     The authors should proofread the manuscript to correct some typos, grammar errors, and unclear expressions. Some examples:

-        Page 4 – Line 77: “in this study” instead of “in this complex”.

-        Page 4 – Line 79: delete “… origin to strong…” instead of “…origin and to strong…”.

-        Page 4 – Line 133: please define immediately “BE” (done only few lines below).

-        Page 11 – Line 316: “extent” instead of “extend”.

-    Page 15 – Lines 406-407: “ammonia-borane complexes” instead of “complex of ammonia borane complexes”.

-        Page 15 – Line 417: please delete “in this group” (repeated twice in the sentence).

-        Page 16 – Line 476: “whose” instead of “which”.

-        Page 16 – Line 490: “whose” instead of “which”.

-     Page 17 – Line 501: “depends on a” instead of “is bases a”. Anyway, the whole sentence (from line 500 to line 502” must be rephrased because not clear.

Author Response

“I found a bit the terms “protonic hydrogen bond” (protonic HB) and “hydridic hydrogen bond” (hydridic HB) misleading. If I understood correctly, these are not hydrogen bonds (in the sense of non-covalent interactions), but bonds that involve hydrogen atoms. In the studied systems, the only non-covalent interactions are the di-hydrogen bonds. I strongly suggest the authors to modify the terms throughout the text as follows (for example): “protonic H-E bond” and “hydridic H-E bond” (where E stands for element). Of course, the authors can propose other alternatives, but not “protonic hydrogen bond” (protonic HB) and “hydridic hydrogen bond” (hydridic HB).

We thank the referee for their very useful suggestion, which has been adapted throughout the revised version of the manuscript.

Table 3 is not discussed at all in the main text. If the authors want to keep it, they should move it to the Supplementary Material.

Table 3 has been moved to the revised Supplementary Material.

Lines 260-261. The authors wrote: “It is noteworthy that some of the hydridic changes in transition metal and noble gas compounds are positive”. By inspecting Table 2, the hydridic charges in noble gas compounds are always negative. On the contrary, those of the aromatic compounds are always positive. The authors should correct this point.

The sentence has been changed in the revised version of the manuscript.

Page 10 – Lines 297-298. The authors wrote: “Furthermore, we also cannot confirm that the protonic part of the complex is more bent whereas the hydridic part is more linear”. Is it not the opposite considering what was discussed few lines above? The authors should correct this sentence or be clearer on this point.”

should correct this point.

The sentence has been changed in the revised version of the manuscript.

“The authors should proofread the manuscript to correct some typos, grammar errors, and unclear expressions.”

Done